# Macrophages from Mice Administered *Rhus verniciflua* Stokes Extract Show Selective Anti-Inflammatory Activity

**DOI:** 10.3390/nu10121926

**Published:** 2018-12-05

**Authors:** Bo-Geun Kim, Youngju Song, Mi-Gi Lee, Jin-Mo Ku, So-Jung Jin, Joung-Woo Hong, SeungGwan Lee, Hee Kang

**Affiliations:** 1Graduate School of East-West Medical Science, Kyung Hee University, Yongin 17104, Korea; rapport6@hanmail.net (B.-G.K.); jwhong46@khu.ac.kr (J.-W.H.); 2Department of Biomedical Science and Technology, Graduate School, Kyung Hee University, Seoul 02447, Korea; songyj@khu.ac.kr; 3Bio Center, Gyeonggido Business and Science Accelerator, Suwon 16229, Korea; migi@gbsa.or.kr (M.-G.L.); medichem@gbsa.or.kr (J.-M.K.); 4Department of Horticultural Biotechnology, College of Life Sciences, Kyung Hee University, Yongin 17104, Korea; truesojung@gmail.com; 5Humanitas College, Kyung Hee University, Yongin 17104, Korea; leesg@khu.ac.kr

**Keywords:** *Rhus verniciflua* Stokes, in vivo, inflammation, macrophage, monocyte differentiation, MHC II, IL-12

## Abstract

The bark of *Rhus verniciflua* Stokes (RVS) is used as a food additive and herbal medicine for various inflammatory disorders and cancer in Eastern Asia. RVS has been shown to exert anti-inflammatory effects in lipopolysaccharide (LPS)-stimulated macrophages in vitro, but whether oral administration of RVS affects the inflammatory response of macrophage needs to be verified. RVS was given orally to mice for ten days. For isolation of macrophages, intraperitoneal injection of thioglycollate was performed. For determination of serum inflammatory response, intraperitoneal injection of LPS was applied. RVS stimulated monocyte differentiation in thioglycollate-induced peritonitis by increasing the population of cells expressing CD11b and class A scavenger receptors. These monocyte-derived macrophages showed an increased uptake of acetylated low-density lipoprotein. When peritoneal macrophages from the RVS group were stimulated with LPS, the levels of tumor necrosis factor (TNF)-α and interleukin (IL)-6 in the supernatant decreased, but the level of IL-12 increased. The surface expression of CD86 was reduced, but surface expression of class II major histocompatibility complex molecules was increased. RVS suppressed the serum levels of LPS-induced TNF-α and IL-6. Collectively, RVS promoted monocyte differentiation upon inflammatory insults and conferred selective anti-inflammatory activity without causing overall inhibitory effects on immune cells.

## 1. Introduction

Monocytes and macrophages are collectively called mononuclear phagocytes, and the function of these cells is to maintain immunity and tissue integrity [1]. Mononuclear phagocytes are involved in phagocytosis, inflammatory response, and the regulation of T cell activity throughout the body [1]. In the past, the consensus was that monocytes and macrophages were produced in the bone marrow, and that tissue macrophages were simply replenished by circulating monocytes. However, recent studies have shown that tissue macrophages are created prior to birth and are maintained through local proliferation, except in some tissues, such as the intestines and dermis, where monocytes constantly migrate [2]. As inflammation progresses, circulating monocytes are recruited to the inflamed tissue and differentiate into inflammatory macrophages [3].

Upon encountering microorganisms, macrophages detect and remove them using various phagocytic receptors, including scavenger receptors [4]. At the same time, macrophages synthesize inducible nitric oxide synthase (iNOS), which converts arginine and oxygen to nitric oxide (NO) and citrulline [5]. NO is cytotoxic against intracellular infectious agents, but excessive production of this gaseous molecule is detrimental to host cells [5]. Activated macrophages secrete inflammatory cytokines such as tumor necrosis factor (TNF)-α, interleukin (IL)-6, and IL-12. TNF-α and IL-6 activate endothelial cells, attract circulating leukocytes to the site, and induce the acute phase response in the liver [6]. IL-12 activates natural killer (NK) cells and enhances the development of type 1 T helper (Th1) cells [7]. Further, such activated macrophages upregulate class II major histocompatibility complex (MHC) molecules and costimulatory molecules to activate antigen-specific Th1 cells, which in turn enhance the macrophage’s function by producing interferon (IFN)-γ, a cytokine once termed “macrophage activation factor” [8]. For these reasons, any strategy for controlling inflammation must include regulation of macrophages, which are the ultimate target of many effective anti-inflammatory agents. 

Thioglycollate is one of the agents that increase peritoneal macrophage yield by inducing sterile peritonitis in mice [9]. These peritoneal macrophages are highly phagocytic but do not produce inflammatory mediators unless they are stimulated with lipopolysaccharide (LPS) or other inflammatory stimuli [9,10]. By three to four days after intraperitoneal injection of thioglycollate, the number of peritoneal exudate cells reaches a peak, and the major population in the peritoneal cavity is monocyte-derived macrophages [9,11]. The process of monocyte-to-macrophage differentiation in vitro is characterized by upregulation of CD11b, class A scavenger receptors (SRAs), and CD36 [12,13,14]. CD11b is a macrophage-specific integrin that mediates the adherence of tissue macrophages to the extracellular matrix [14]. CD36 and SRAs function as scavenger receptors that recognize polyanionic macromolecules [15]. Peritoneal exudate cells from thioglycollate-injected mice show gradual upregulation of CD11b and CD36 with time [16]. In this regard, thioglycollate-induced peritonitis can be an in vivo model for evaluating monocyte-to-macrophage differentiation [16]. 

The bark of *Rhus verniciflua* Stokes (RVS), or *Toxicodendron vernicifluum* (Stokes) F. Barkley, which belongs to the Anacardiaceae family, is used as a food additive and as herbal medicine for inflammatory disorders and cancer in Eastern Asia. RVS demonstrates various biological activities, including anti-oxidant, anti-inflammatory, neuroprotective, and anti-tumor action—some of which have been verified in vitro [17,18,19,20,21,22,23,24]. At the cellular level, RVS extract and its phenol-rich fraction inhibit lipopolysaccharide (LPS)-induced TNF-α and IL-6 expression in RAW264.7 cells, a mouse macrophage cell line [18,19,20,24]. Butein, sulfuretin, and fisetin are the known anti-inflammatory polyphenols present in RVS [25,26,27]. Animals orally administered RVS extract exhibit reduced edema in chemically-induced dermatitis models [19,24], but whether this anti-inflammatory effect occurs in macrophages following oral administration of RVS extract has not been determined. A pharmacokinetic profile study showed that the majority of fisetin, sulfuretin, and other polyphenols in RVS extract were present in blood in their conjugated forms [28]. This indicates that inferences drawn from the in vitro anti-inflammatory activity of RVS extract may not be relevant to in vivo conditions. 

We aimed to ascertain whether macrophages isolated from mice orally given RVS extract showed anti-inflammatory activity similar to that seen in macrophages directly treated with RVS extract or its polyphenol fractions. For this purpose, we used the thioglycollate-induced peritonitis model to obtain a high yield of macrophages and to evaluate whether RVS extract affected monocyte differentiation. In addition, we investigated the effects of RVS extract on macrophage function relating to Th cell activity. Finally, we examined the systemic response to LPS in mice administered RVS extract.

## 2. Materials and Methods

### 2.1. Sample Preparation

Ten-year-old RVS bark was collected in Jecheon, Chungbuk Province, South Korea, and was dried and chopped. A voucher specimen (# 2013-RVS) was deposited at the laboratory of herbal immunology, Kyung Hee University. Because RVS contains urushiol, an allergen that can cause contact dermatitis, the removal of urushiol was performed as described by Choi et al [23]. In brief, the RVS bark pieces were roasted in an iron pot at 240 °C for 50 min. They were then extracted in water using a heating mantle and reflux apparatus for 2 h at 95 °C and filtered through Whatman number 2 filter paper (Whatman International, Kent, England). The extract was concentrated using a rotary evaporator and freeze-dried under vacuum. The yield was 2.25%. 

### 2.2. Qualitative and Quantitative Analyses of RVS Extract

High performance liquid chromatography combined with mass spectrometry was used to analyze the amount and identity of the compounds in the RVS extract. The monoisotopic masses of protonated fisetin ([M + H]^+^, 287.1) and sulfuretin ([M + H]^+^, 271.1) were detected with sufficient resolution. The integrals of mass using selected ion monitoring showed the amount of the compounds in the RVS extract. The quantitative analysis was performed by comparison with commercially available fisetin and sulfuretin after calibration curves of those compounds were obtained with a reliable coefficient of determination (*R*^2^ > 0.996). The liquid chromatography was performed using a Waters ACQUITY UPLC BEH C18 column (2.1 × 100 mm; Milford, MA, USA). Mobile phase A was 0.1% formic acid in water, and phase B was 0.1% formic acid in acetonitrile. The mass spectrometry was performed using a Micromass Quattro Micro API mass spectrometer (Milford, MA, USA). The amounts of fisetin and sulfuretin in the RVS extract are presented in Table 1. 

### 2.3. Animals

Seven-week-old male BALB/c mice were obtained from Samtako (Osan, Korea) and housed in a temperature- and humidity-controlled pathogen-free animal facility with a 12 h light-dark cycle. The animal protocol was approved by the Institutional Animal Care and Use Committee at Kyung Hee University (KHUASP(SE)-15-012), and the mice were cared for according to the US National Research Council for the Care and Use of Laboratory Animals (1996) specifications. All animals underwent 1 week of adjustment prior to experiments and were randomly allocated to the experimental groups. RVS extract dissolved in water (20 mg/mL or 100 mg/mL) was given orally to mice at doses of 200 mg/kg or 1000 mg/kg body weight once daily for 10 days. Control mice were given an equal volume of water. Mice were assigned for two experiments: peritoneal macrophage isolation (*N* = 6) and intraperitoneal injection of LPS (*N* = 12). These procedures are given below. 

### 2.4. In Vivo Experiment for Peritoneal Exudate Cell Preparation

On day 6 after the first RVS extract treatment, control and RVS-treated mice were injected intraperitoneally with 2 mL of 3.5% sterile thioglycollate (BD, Sparks, MD, USA). On day 10, thioglycollate-injected mice were sacrificed via cervical dislocation 1 h after the final RVS extract treatment. The peritoneal exudate cells were aseptically isolated via peritoneal lavage with cold Dulbecco’s Modified Eagle’s Medium (DMEM) (HyClone, Logan, UT, USA) containing 10% fetal bovine serum (FBS; HyClone, Logan, UT, USA) and 1% penicillin-streptomycin (WEL GENE, KyungSan, Korea). After centrifugation, the cells were resuspended and counted using a TC20 Cell Counter (Bio-Rad Laboratories, Hercules, CA, USA). Then some peritoneal exudate cells were evaluated for the identification of CD11b and scavenger receptors by flow cytometry. The remaining cells were used for cell culture. 

### 2.5. Peritoneal Macrophage Culture

Peritoneal exudate cells from control and RVS groups were plated in six-well plates overnight at 37 °C, and the non-adherent cells were removed. For scavenger receptor activity, the cells were cultured with Alexa Fluor 488-labeled acetylated low-density lipoprotein (acLDL) (Molecular Probes, Eugene, OR, USA) for 4 h. Cells were washed with phosphate buffered saline (PBS) and detached by scraping. Then cells were subjected to flow cytometric analysis. 

In order to quantify cytokines and costimulation molecules produced by peritoneal macrophages, the cells were stimulated with 100 ng/mL LPS (serotype 055: B5, St. Louis, Sigma, MO, USA) for 24 h. The supernatant was collected for enzyme-linked immunosorbent assay (ELISA). The cells were washed with PBS and detached by scraping. Cells were then subjected to flow cytometric analysis.

### 2.6. RAW264.7 Cell Culture

The mouse macrophage cell line RAW264.7 cells were purchased from the Korea Cell Line Bank (Seoul, Korea) and maintained in DMEM with 10% FBS and 1% penicillin-streptomycin. In order to compare the in vitro effect of RVS extract on cytokine production, RAW264.7 cells were plated in 24-well plates and stimulated with 100 ng/mL LPS in the presence of RVS extract for 24 h and supernatant was collected for ELISA. 

### 2.7. Flow Cytometry

For the measurement of surface molecules, the cells were washed twice in PBS and resuspended at 1 × 10^6^ cells/mL in FACS buffer (PBS/0.1% NaN_3_/1% FBS). The cells were blocked with rat anti-mouse CD16/CD32 antibody (BD Biosciences, San Diego, CA, USA) at 4 °C for 5 min and then stained with fluorescein-conjugated anti-mouse SR-AI (R&D Systems, Minneapolis, MN, USA), phycoerythrin (PE)-conjugated anti-mouse CD36, fluorescein isothiocyanate (FITC)-conjugated CD11b, PE-conjugated anti-CD11b, PE-conjugated anti-mouse CD86, or FITC-conjugated anti-mouse MHC II (IA/IE) (BD Biosciences, San Diego, CA, USA) for 30 min on ice in the dark. Matched isotype antibodies were used to show non-specific binding. The cells were washed and resuspended in FACS buffer. A total of 10,000 events were acquired using a Navios flow cytometer (Beckman Coulter, La Brea, CA, USA), and the data were processed using Kaluza software, version 1.2 (Beckman Coulter, La Brea, CA, USA). For the measurement of intracellular uptake of Alexa Fluor 488-labeled acLDL, harvested cells were washed twice in PBS and resuspended at 1 × 10^6^ cells/mL in FACS buffer. Then cells were subjected to flow cytometry.

### 2.8. In Vivo Experiment for Serum Cytokine Response

One hour after the final RVS extract treatment on day 10, mice were intraperitoneally injected with 1.3 mg/kg of LPS (serotype 055: B5, Sigma, St. Louis, MO, USA). After 1 h, the mice were anesthetized with ether, and blood was collected via cardiac puncture. Serum was obtained for measurement of cytokine analysis.

### 2.9. ELISA

The levels of TNF-α, IL-6, IL-12p70, and IFN-γ in the supernatants or sera were determined using BD OptEIA mouse ELISA sets (BD Biosciences, San Diego, CA, USA) according to the manufacturer’s protocol.

### 2.10. Statistical Analysis

Tests of normality were performed using the Kolmogorov-Smirnov test. Data were presented as mean ± standard error of the mean (SEM). Two-sided Student’s *t*-test or one-way analysis of variance was applied to compare the differences between groups. If the statistical analysis showed that the differences between multiple groups were significant, Tukey’s test was used for further comparison. All statistical analyses were performed using SPSS software, version 22.0 (IBM, Chicago, IL, USA). For this study, *p*-values less than 0.05 were considered statistically significant. 

## 3. Results

### 3.1. Effects of RVS Extract on Thioglycollate-Induced Monocyte Differentiation in Peritoneal Exudate Cells

We first investigated whether RVS extract affects monocyte-to-macrophage differentiation in the thioglycollate-peritonitis model. RVS extract was orally given to mice for 10 consecutive days. There were no differences in body weight among the groups during the experimental period. One hour after the final RVS extract administration, mice were sacrificed for peritoneal cell isolation. Peritoneal exudate cells consist of various cell populations. We used CD11b as a marker for macrophages and defined cells expressing CD11b and scavenger receptors CD36 and SRAs as monocyte-derived macrophages using flow cytometry. There was no difference in cells expressing both CD11b and CD36 between the control and RVS groups (Figure 1A,C). The proportion of SRAs(+)CD11b(+) cells increased in the 200 mg/kg RVS group (Figure 1B,D). These data show that RVS extract treatment enhances thioglycollate-induced monocyte differentiation in the peritoneum. 

### 3.2. Effect of RVS Extract on Scavenger Receptor Activity in Peritoneal Macrophages

We further examined the activity of macrophage scavenger receptors. Peritoneal exudate cells from the control and RVS-treated mice were incubated overnight and non-adherent cells were removed. In our laboratory, we routinely find over 90% of adherent peritoneal exudate cells to be CD11b (+). We cultured these adherent peritoneal macrophages with Alexa Fluor 488-labeled acLDL, a ligand for scavenger receptors, and determined the intracellular uptake of the labeled acLDL by quantifying mean fluorescence intensity. A dose-dependent increase in Alexa Flour 488-labeled acLDL uptake was observed in the RVS extract group (Figure 2). These results indicate that RVS extract enhances the activity of scavenger receptors expressed by monocyte-derived peritoneal macrophages. 

### 3.3. Effects of RVS Extract on TNF-α and IL-6 Production in Peritoneal Macrophages Stimulated with LPS

Peritoneal macrophages from the control and RVS extract groups were stimulated with LPS for 24 h, and the levels of secreted TNF-α and IL-6 were measured by ELISA. A significant reduction in TNF-α production was noted in cells from the 200 mg/kg group (Figure 3A). The levels of IL-6 decreased in a dose-dependent manner (Figure 3B). In comparison with these findings, we treated RAW264.7 cells with RVS extract and stimulated these cells with LPS for 24 h. We found that concentrations of RVS up to 200 µg/mL were not cytotoxic using the MTT assay. In our in vitro experiments, RVS did not affect the levels of LPS-induced TNF-α but decreased those of IL-6, as measured by ELISA (Figure 4A,B).

### 3.4. Effects of RVS Extract on CD86 and Class II MHC Surface Expression in Peritoneal Macrophages Stimulated with LPS

We investigated whether RVS extract affected the surface expression of CD86 and class II MHC molecules, which link macrophages to Th cells. Significant changes were noted in the 1000 mg/kg RVS group. The mean fluorescence intensity of CD86 decreased (Figure 5A,C) and that of class II MHC molecules increased (Figure 5B,D). These data show that the higher dose of RVS extract induces a differential response in surface molecules relating to Th cells. 

### 3.5. Effects of RVS Extract on IL-12 and IFN-γ Production in Peritoneal Macrophages Stimulated with LPS

We investigated whether RVS extract altered IL-12 response in peritoneal macrophages stimulated with LPS. In contrast to TNF-α and IL-6, increases in LPS-induced IL-12 production were observed in macrophages from the RVS extract-treated groups (Figure 6A). Macrophages are not the dominant source of IFN-γ, but IL-12 has been reported to induce the secretion of IFN-γ in peritoneal macrophages [29]. We measured the level of IFN-γ in the supernatants of the activated macrophage cultures but found no difference in the control and RVS groups (Figure 6B). RVS extract differentially regulated the production of macrophage-derived cytokines in response to LPS.

### 3.6. Effects of RVS Extract on the Systemic Response to LPS

We tested whether the systemic cytokine response after LPS challenge is altered in RVS extract-treated mice. Intraperitoneal injection of a sublethal dose of LPS was performed on mice that had been administered 1000 mg/kg of RVS extract. Since the peak serum TNF-α level occurs approximately 1 h after LPS challenge, we obtained serum at this time point [30]. The levels of serum TNF-α and IL-6 in the RVS group were decreased by 63% and 45%, respectively (Figure 7). Because of the delayed production of serum IL-12 and IFN-γ in comparison to TNF-α, we were not able to detect those cytokines. 

## 4. Discussion

Previous studies have shown that in vitro treatment of macrophages with RVS extract or its active components suppresses LPS-induced inflammatory mediators [18,20,24]. Here, we demonstrated that macrophages isolated from mice orally given RVS extract showed a similar activity to that seen in vitro. Further, we discovered that RVS extract affected monocyte differentiation by increasing the number of mature peritoneal macrophages and their scavenger receptor activity and modulated the interaction between macrophages and Th cells.

In the normal mouse peritoneum, the major cell types are resident macrophages and B-1 cells, with the remaining minor groups consisting of T cells, NK cells, NKT cells, and other immune cells [9,11]. Following intraperitoneal injection of thioglycollate, neutrophils are initially recruited, and then monocyte-derived macrophages become the predominant cell type in the peritoneum [9,11,16]. In thioglycollate-induced peritonitis, the number of eosinophils is also increased [9]. Because eosinophils express a low level of CD11b [11], we defined macrophages as cells that express both CD11b and scavenger receptors. Scavenger receptors were originally discovered based on their ability to detect and remove acetylated lipoprotein [15]. Their major roles are to clear pathogens and endogenous waste products, aiding in phagocytosis, antigen presentation, and clearance of apoptotic cells by macrophages [15]. We found over 70% of peritoneal exudate cells obtained four days after thioglycollate injection to be either CD11b(+)CD36(+) cells or CD11b(+)SRAs(+) cells. Mice orally given RVS extract showed an increased population of CD11b(+)SRAs(+) cells in response to thioglycollate and a concurrent increase in uptake of acLDL by peritoneal macrophages. These results suggest that RVS extract may promote the process of monocyte-to-macrophage differentiation and the clearance of unwanted products upon inflammatory insults. 

Peritoneal macrophages from the RVS-treated mice showed reduced secretion of TNF-α and IL-6 after stimulation with LPS. In contrast, RVS extract decreased the secretion of IL-6 but not that of TNF-α in RAW264.7 cells stimulated by LPS in our in vitro system. Since the RVS extract used here was prepared in boiling water, a significant amount of polysaccharide was included in the extract. It seems that incubation of macrophages with polysaccharides can stimulate the release of TNF-α protein, despite their inhibitory effects on TNF-α gene [31]. Importantly, such difference in our in vitro and in vivo findings raises a question as to the evaluation of natural products. When orally consumed, polysaccharides or glycosides can be metabolized by intestinal bacteria. Furthermore, polyphenols are present in their conjugated form after oral administration of RVS in rats [28]. These factors must be taken into account when attempting to extrapolate an in vitro effect of natural products to the in vivo situation.

Human volunteers following intravenous administration of endotoxin show elevated levels of plasma TNF-α accompanied by fever and tachycardia [32]. Experimental studies using rodents demonstrate that the liver and peritoneal macrophages are the major sources of early serum TNF-α in response to intraperitoneal LPS [33,34]. We found that LPS-induced serum levels of TNF-α and IL-6 were remarkably suppressed in the RVS extract group, even more so than in the supernatants of activated macrophages isolated from RVS extract-treated mice. This is likely due to the fact that macrophages are continuously exposed to LPS in vitro, but LPS is rapidly cleared by the liver in vivo [35]. A similar study in which rats were orally given RVS extract at 100 mg/kg and 500 mg/kg for 2 weeks and then challenged with intraperitoneal LPS showed a significant reduction in IL-6 protein and a marginal reduction in TNF-α protein in the liver [36]. These hepatic cytokine concentrations were measured 16 h after LPS stimulation [36]. The peak response of serum TNF-α is approximately 1 h after intraperitoneal challenge with LPS and rapidly declines [30]. Because of the rapid kinetic response of TNF-α, the previous study may have failed to detect a dramatic reduction in the liver. Considering the above findings and ours, the decrease in serum inflammatory response by RVS extract is likely to be mediated by its modulation of hepatic and peritoneal macrophages.

CD86 has increased expression in macrophages stimulated with pathogenic microbes and enhances T cell activity by binding to CD28 on T cells [37]. Class II MHC molecules are required to present antigens to Th cells [8]. Interestingly, macrophages from the 1000 mg/kg RVS extract group showed increased expression of class II MHC and decreased expression of CD86. Expression of class II MHC is increased by IFN-γ [38]. A similar, though not statistically significant, increase in IFN-γ was found in the supernatants of activated macrophages from the 1000 mg/kg RVS extract group, which may have driven the increase in class II MHC expression. Such differential regulation of CD86 and class II MHC indicates that RVS extract may enhance the capacity of macrophages to present antigens to Th cells but reduce their stimulatory signal to activate Th cells. This property may be beneficial for a prolonged immune response involving macrophages and Th cells together. 

IL-12 is produced by macrophages and other phagocytes stimulated by microorganism and their products and is a potent inducer of IFN-γ production in NK cells and Th1 cells [7]. IL-12 plays a critical role in intracellular pathogen control and anti-tumor immunity [7,39]. IL-12p70 is the active form of IL-12 consisting of p35 and p40, which are regulated by different chromosomes [40]. The production of IL-12p35, and therefore of IL-12, requires NF-κB and IRF-1, the latter of which is activated by IFN-γ [41]. However, the cellular origin of IFN-γ in the culture of peritoneal macrophages is unclear [42]. Adherent peritoneal exudate cells contain small quantities of T cells, NK cells, and NKT cells, which produce a large amount of IFN-γ in response to LPS. Therefore, these lymphoid cells could be the main source of the IFN-γ detected in the activated macrophage culture. Whatever the source of the IFN-γ, RVS administration may stimulate macrophages to respond to IFN-γ and produce more IL-12. Because IL-12 is not produced by macrophage cell lines, previous studies on the anti-inflammatory activity of RVS extract using RAW265.7 cells did not detect an IL-12 response. This report sheds light on the ability of RVS extract to stimulate IL-12 response in activated macrophages. 

Macrophages detect LPS through toll-like receptor 4, which then initiates the induction of inflammatory and immune responses by using MyD88 and TRIF as adaptors to transmit signals to the nucleus [43]. The MyD88-depenent pathway activates NF-κB and mitogen-activated protein kinase (MAPK) to induce the expression of inflammatory proteins such as TNF-α, IL-6, and iNOS [44]. The TRIF-dependent pathway is involved in inflammatory gene expression and, more importantly, is indispensable in the upregulation of costimulatory molecules, such as CD86, and class II MHC molecules that bridge macrophages to Th cells [45]. In vitro, RVS contains some active components that are able to inhibit NF-κB and MAPK signaling in macrophages [18]. Our findings show that macrophages from the RVS extract-treated mice had decreased levels of TNF-α, IL-6, and CD86 and increased levels of IL-12 and class II MHC molecules, indicating that unidentified compounds found in RVS extract or their metabolites may exert multiple effects on these signaling pathways. The detailed mechanism that may account for the action of RVS extract needs to be further verified. 

## 5. Conclusions

Taken together, we demonstrated for the first time that oral administration of RVS extract in mice enhanced monocyte differentiation by increasing the population of mature macrophages and their scavenger receptor activity in the thioglycollate-induced peritonitis model. When these monocyte-derived macrophages from mice given RVS extract were stimulated with LPS, they showed reduced production of TNF-α and IL-6 and reduced surface expression of CD86 but increased surface expression of class II MHC and increased production of IL-12. Although detailed mechanistic studies on such differential effects are needed, these findings support the hypothesis that RVS confers anti-inflammatory activity without causing overall inhibitory effects on immune cells.

## Figures and Tables

**Figure 1 nutrients-10-01926-f001:**
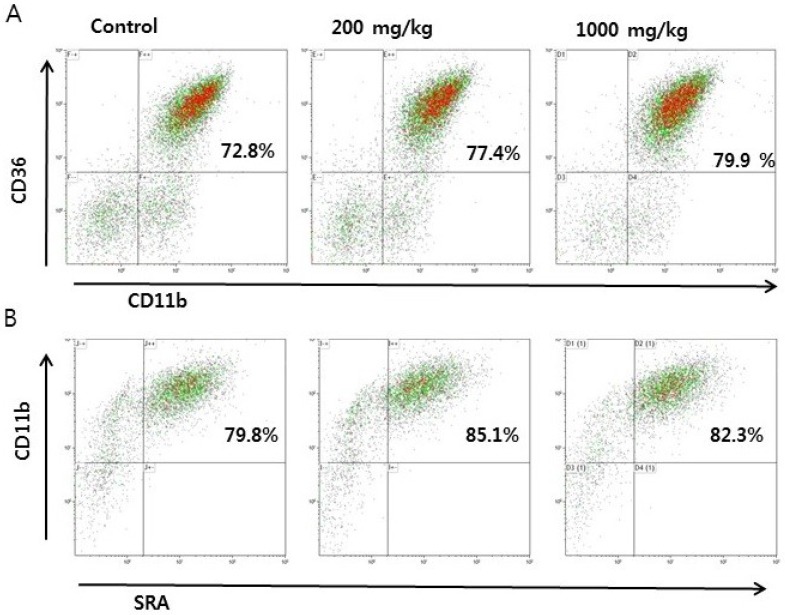
Effects of *Rhus verniciflua* Stokes (RVS) extract on the percentage of monocyte-derived macrophages in peritoneal exudate cells. Mice were orally given RVS extract (200 or 1000 mg/kg) for 10 days. On day 10, peritoneal exudate cells were obtained following the final administration of RVS. Intraperitoneal injection of thioglycollate was performed 4 days before the isolation of peritoneal exudate cells. (**A**,**B**) Peritoneal exudate cells were double-stained with FITC-labeled anti-CD11b and PE-labeled anti-CD36 antibodies (**A**) or FITC-labeled anti-SRA and PE-labeled anti-CD11b antibodies (**B**) and analyzed via flow cytometry. Representative dot plots are shown. (**C**,**D**) The bars represent the mean ± standard error of the mean (SEM) (*N* = 6). Statistical differences between groups were determined using Student’s *t*-test. * *p* < 0.05 vs. control.

**Figure 2 nutrients-10-01926-f002:**
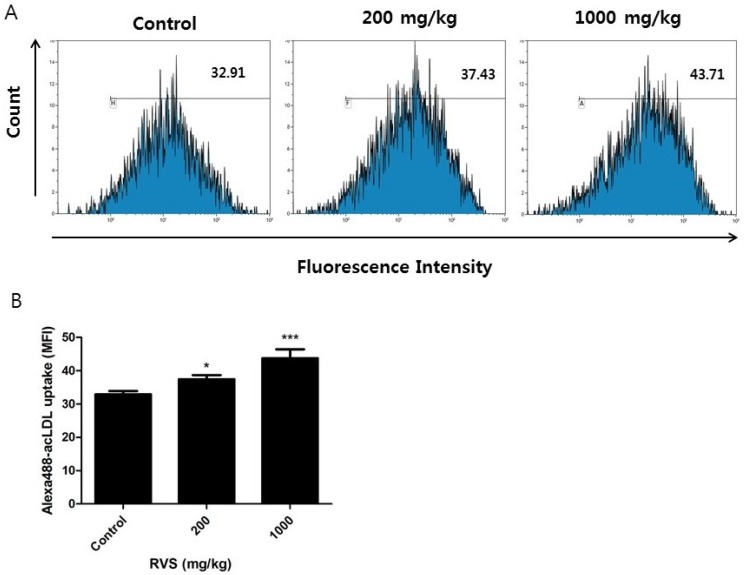
Effects of RVS extract on scavenger receptor activity in peritoneal macrophages. Peritoneal exudate cells isolated from thioglycollate-injected mice were cultured overnight. After removal of non-adherent cells, the remaining adherent cells were incubated with Alexa Fluor 488-labeled labeled acetylated low-density lipoprotein (acLDL) for 4 h and analyzed via flow cytometry. (**A**) Representative histograms are shown. (**B**) The bars represent the mean ± SEM (*N* = 6). MFI: mean fluorescence intensity. Statistical differences between groups were determined using Student’s *t*-test. * *p* < 0.05, *** *p* < 0.005 vs. control.

**Figure 3 nutrients-10-01926-f003:**
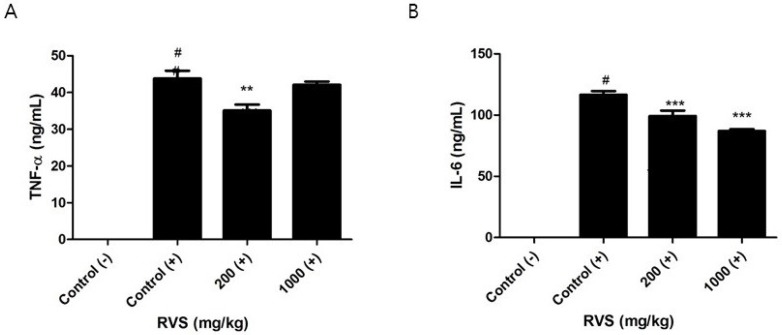
Effects of RVS extract on tumor necrosis factor (TNF)-α and interleukin (IL)-6 levels in the supernatants of lipopolysaccharide (LPS)-stimulated peritoneal macrophages. Adherent peritoneal macrophages isolated from mice orally given RVS extract were stimulated with LPS for 24 h, and the concentrations of TNF-α (**A**) and IL-6 (**B**) were measured using enzyme-linked immunosorbent assay (ELISA). The bars represent the mean ± SEM (*N* = 6). (−): without LPS, (+): with LPS. Statistical differences between groups were determined using one-way ANOVA with Tukey’s test. # *p* < 0.005 vs. control (−). ** *p* < 0.01, *** *p* < 0.005 vs. control (+).

**Figure 4 nutrients-10-01926-f004:**
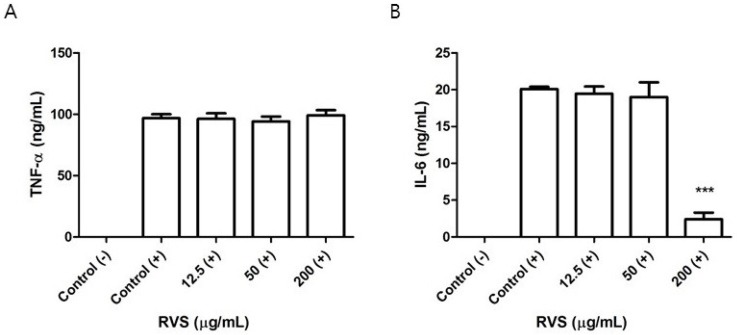
Effects of RVS extract on TNF-α and IL-6 levels in the supernatants of LPS-stimulated RAW264.7 cells. Cells were stimulated with LPS in the presence of RVS extract for 24 h, and the concentrations of TNF-α (**A**) and IL-6 (**B**) were measured using ELISA. The bars represent the mean ± SD (*N* = 3). (−): without LPS, (+): with LPS. Statistical differences between LPS-treated control cells and LPS/RVS-treated cells were determined using Student’s *t*-test. *** *p* < 0.005 vs. control (+).

**Figure 5 nutrients-10-01926-f005:**
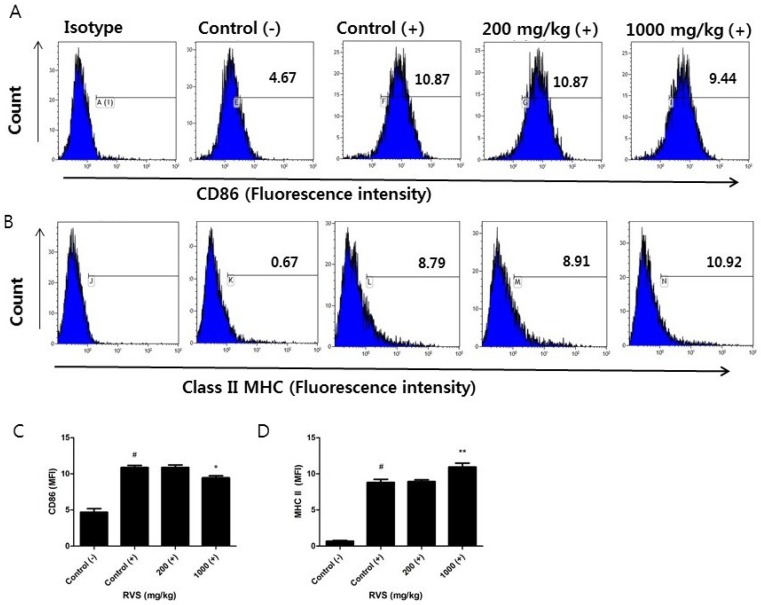
Effects of RVS extract on surface expression of CD86 and class II MHC in LPS-stimulated peritoneal macrophages. Adherent peritoneal macrophages from the control and RVS-treated mice were stimulated with LPS for 24 h and surface expression of CD86 and class II MHC molecules was determined via flow cytometry. (**A**,**B**) Representative histograms are shown. (**C**,**D**) The bars represent the mean ± SEM (*N* = 6). (−): without LPS, (+): with LPS. Statistical differences between groups were determined using one-way ANOVA with Tukey’s test. # *p* < 0.005 vs. control (−). * *p* < 0.05, ** *p* < 0.01 vs. control (+).

**Figure 6 nutrients-10-01926-f006:**
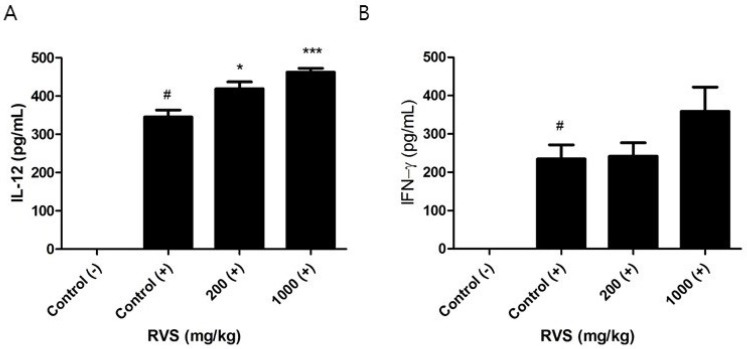
Effects of RVS extract on IL-12 and IFN-γ levels in the supernatants of LPS-stimulated peritoneal macrophages. Adherent peritoneal macrophages were stimulated with LPS for 24 h, and the concentrations of IL-12 (**A**) and IFN-γ (**B**) were measured. The bars represent the mean ± SEM (*N* = 6). Statistical differences between groups were determined using one-way ANOVA with Tukey’s test. (−): without LPS, (+): with LPS. # *p* < 0.005 vs. control (−). * *p* < 0.5, *** *p* < 0.005 vs. control (+).

**Figure 7 nutrients-10-01926-f007:**
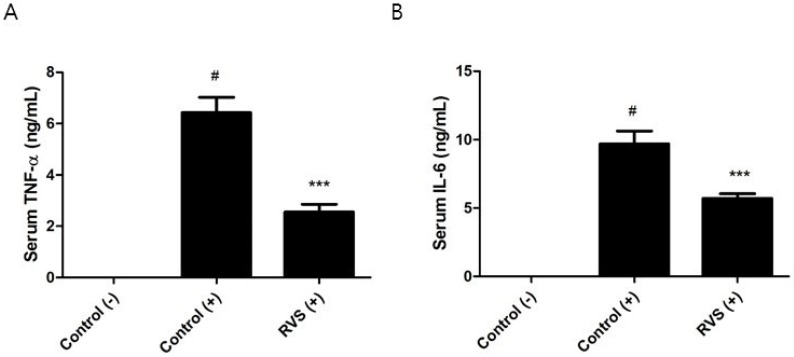
Effects of RVS extract on the systemic response to LPS. Following oral administration of RVS extract, mice were intraperitoneally injected with a sublethal dose of LPS (1.3 mg/kg), and serum was obtained one hour later. The levels of TNF-α (**A**) and IL-6 (**B**) were measured using ELISA. The bars represent the mean ± SEM (*N* = 10). (−): without LPS, (+): with LPS. Statistical differences between groups were determined using one-way ANOVA with Tukey’s test. # *p* < 0.005 vs. control (−). *** *p* < 0.005 vs. control (+).

**Table 1 nutrients-10-01926-t001:** Quantitative analysis of biologically active compounds in *Rhus verniciflua* Stokes (RVS) extract.

Compounds	MS [M + H]^+^	Content (%)
Fisetin	287.1	0.56
Sulfuretin	271.1	0.01

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
