# Peer review of "Macrophages from Mice Administered Rhus verniciflua Stokes Extract Show Selective Anti-Inflammatory Activity"

_nutrients, 2018, doi:10.3390/nu10121926_

Reviewer 1 Report

In their manuscript Kim and Colleagues investigate the effect of Rhus verniciflua Stokes (RVS) on macrophages and their response to inflammatory stimuli. This is an interesting study yet preliminary. Data are well presented but this study warrant further investigations particularly in vivo in order to make of that study an improvement compared to other studies investigating RVS effect on macrophages in vitro. Understanding further in what macrophages from RVS treated mice could be less inflammatory would be ideal. Some other specifics comments are detailed below:

-        While it was described in other studies, showing the effect of RVS on Raw-264.7 in the present study will allow to compare the production of TNFa, IL-6 in RVS-treated peritoneal macrophages and in RVS-treated raw in your experimental conditions.

-        Overall, the RVS effects remain mild in the in vitro cultured peritoneal macrophages and sometime do not reach significance. Are these effects relevant? In what can mild effect can still have a physiological relevance?

-        In vivo data presented figure 6 are more convincing than the in vitro ones and would warrant further investigation. Is the decreased LPS-induced systemic inflammation in RVS treated mice dependent on the activation of the peritoneal macrophages? That question should be elucidated in order to provide a better rational to this study and the results presented in figure 1-5.

Author Response

Q1  While it was described in other studies, showing the effect of RVS on Raw-264.7 in the present study will allow to compare the production of TNFa, IL-6 in RVS-treated peritoneal macrophages and in RVS-treated raw in your experimental conditions.

R1) As recommended by Reviewer, we performed in vitro evaluation. We treated RAW264.7 cells or peritoneal macrophages with RVS extract and stimulated these cells with LPS for 24 h. RAW264.7 cells and peritoneal macrophages showed the same pattern. In vitro, RVS extract decreased the secretion of IL-6 but not that of TNF-α in LPS-stimulated macrophages (we only presented the results with RAW264.7 cells in Figure 4). These findings are not consistent with those previously reported. We think that this difference is caused by the presence of polysaccharides in our RVS extract. Previous studies in which the polyphenol-rich fraction of RVS, which contained lower levels of polysaccharides was evaluated showed a reduction in secreted TNF-α and IL-6 in LPS-stimulated RAW264.7 cells. The RVS extract used in our manuscript was prepared in boiling water without further processing. Some polysaccharides such as fucoidan can bind macrophage scavenger receptors and induce TNF-α production without alteration of its mRNA (please refer to reference 36 in the revision). Park et al. used hot water extraction similar to that used by us and reported the inhibitory effect of RVS on TNF-α and IL-6 gene expression in LPS-stimulated RAW264.7 cells (please refer to reference 24 in the revision). TNF-α secretion can be regulated at the transcriptional and posttranscriptional level (please refer to reference 37 in the revision). In our in vitro condition, incubation of macrophages with RVS extract containing polysaccharides must have stimulated the release of TNF-α protein despite its inhibitory effects on TNF-α gene. These statements were added in Discussion section (please see lines 312 to 327).

Q2)  Overall, the RVS effects remain mild in the in vitro cultured peritoneal macrophages and sometime do not reach significance. Are these effects relevant? In what can mild effect can still have a physiological relevance?

R2) Reviewer is right that the effect of oral administration of RVS on the inflammatory response of macrophages is mild, compared to its effect on the serum inflammatory response. We think that part of the reason may be due to the difference between in vitro and in vivo experimental conditions. In vitro culture, macrophages are exposed to LPS for 24 h and TNF-α is accumulating in the media, exerting a positive feedback on macrophages. In vivo, LPS is cleared by the liver and TNF-α is consumed not only by macrophages but also by other cells. This statement is included in Discussion (lines 331-334). Although our in vitro experiments are not close to in vivo situations, the physiological relevance of our findings is that macrophages after oral administration of RVS tend to be less inflammatory. We hope that our interpretation will meet the reviewer’s concern.

Q3)  In vivo data presented figure 6 are more convincing than the in vitro ones and would warrant further investigation. Is the decreased LPS-induced systemic inflammation in RVS treated mice dependent on the activation of the peritoneal macrophages? That question should be elucidated in order to provide a better rational to this study and the results presented in figure 1-5.

R3) Thank you for giving us the opportunity to gain a better insight into our findings. Experimental studies using rodents show that the liver and peritoneal cells are the major sources of early serum TNF-α peak. Rats with a two-thirds hepatectomy produced 64% less TNF-α in serum following an endotoxin injection (please refer to reference 39 in the revision). Mice that had been eliminated of peritoneal cells showed a reduction by 55% in serum TNF-α (please refer to reference 42 in the revision). From these studies, it is apparent that the early elevation of serum TNF-α comes from the liver macrophages and peritoneal macrophages. A previous study showed that oral administration of RVS decreased the concentrations of IL-6 and TNF-α protein and the mRNA expression of iNOS in the liver of rats at 16 h after LPS injection (please refer to reference 36 in the revision Moon 2015). Based on their findings and ours, we assume that the inhibitory effects of RVS on serum cytokines following intraperitoneal injection of LPS are likely to be mediated by its modulation of hepatic and peritoneal macrophages. We added these statements in Discussion (lines 328-342).

Reviewer 2 Report

The paper by Bo-Geun Kim et al. report the anti-inflammatory effect of orally administered extract of Rhus vernicifluaStokes on mouse peritoneal macrophages. The manuscript is well written and the experimental design is accurate. This is a well conducted study that confirm previous results showing anti-inflammatory effect of this plant extract, especially phenolic extract. Although the main results, i.e. inhibition of IL-6 and TNF-asecretion and reduced CD86 expression, have been already reported in murine macrophage cell line RAW-264.7, it was important to confirm this in vivo, especially after oral administration of RVS extract. In addition, the authors also showed increasing expression of scavenger receptors and that macrophages from RVS fed mice respond differently to LPS stimulation by an increase of IL-12 and MHC-II proteins. However, it is surprising that nitric oxide (NO) synthesis, which was reported to be inhibited by RVS extract in RAW-264.7 cells, was not investigated in this paper. Since the conclusive hypothesis of the author is that RVS may selectively affect the NF-kB pathway, it seems relevant to investigate the NO production, which is dependent of NF-kB, in LPS-stimulated macrophages after RVS administration.

At least, authors should add some elements about NO synthesis in the introduction and/or discussion. Further, adding experiments to show the inhibition of NO synthesis and NO synthase expression would add strength to the hypothesis of NF-kB pathway as a target of RVS extracts.

Author Response

The paper by Bo-Geun Kim et al. report the anti-inflammatory effect of orally administered extract of Rhus vernicifluaStokes on mouse peritoneal macrophages. The manuscript is well written and the experimental design is accurate. This is a well conducted study that confirm previous results showing anti-inflammatory effect of this plant extract, especially phenolic extract. Although the main results, i.e. inhibition of IL-6 and TNF-asecretion and reduced CD86 expression, have been already reported in murine macrophage cell line RAW-264.7, it was important to confirm this in vivo, especially after oral administration of RVS extract. In addition, the authors also showed increasing expression of scavenger receptors and that macrophages from RVS fed mice respond differently to LPS stimulation by an increase of IL-12 and MHC-II proteins. However, it is surprising that nitric oxide (NO) synthesis, which was reported to be inhibited by RVS extract in RAW-264.7 cells, was not investigated in this paper. Since the conclusive hypothesis of the author is that RVS may selectively affect the NF-kB pathway, it seems relevant to investigate the NO production, which is dependent of NF-kB, in LPS-stimulated macrophages after RVS administration.

At least, authors should add some elements about NO synthesis in the introduction and/or discussion. Further, adding experiments to show the inhibition of NO synthesis and NO synthase expression would add strength to the hypothesis of NF-kB pathway as a target of RVS extracts.

Response ) As suggested by Reviewer, we added some comments on NO in the introduction (lines 48-51). Reviewer is right that we should investigate NO production in macrophages obtained from mice orally given RVS to provide evidence of NF-kB modulation. We failed to detect NO release in the supernatant of LPS-stimulated macrophages. We tried to solve this problem through literature search. As described in Materials and methods, the mouse strain we used was BALB/c, a Th2 type strain. Mills et al reported that the NO response of Balb/c macrophages in response to LPS varies. In fact, we observe the same phenomenon(please refer to reference 51 in the revision). These authors discovered that BALB/c macrophages produce more TGFb than C57BL/6 macrophages do. TGFb inhibits iNOS expression, which explains why BALB/c macrophages show a weak NO response. BALB/c macrophages require IFN-gamma to produce NO. Although we detected IFN-gamma in the supernatant of LPS-stimulated macrophages, it must have been insufficient to activate peritoneal macrophages to produce detectable amounts of NO. We discussed our failure to detect NO in the end of Discussion (lines 374-382). In the future, we will be planning to use C57BL/6 mice to investigate the effect of RVS on NO and other NF-kB dependent genes.  

Round  2

Reviewer 1 Report

The authors adequately addressed my comments. I have no additional concerns. 

Author Response

Thanks so much for your positive feedback, we appreciate your valuable comments.